# Associations between Meteorological Factors and Visceral Leishmaniasis Outbreaks in Jiashi County, Xinjiang Uygur Autonomous Region, China, 2005–2015

**DOI:** 10.3390/ijerph16101775

**Published:** 2019-05-20

**Authors:** Yi Li, Canjun Zheng

**Affiliations:** 1State Key Laboratory of Severe Weather & Key Laboratory of Atmospheric Chemistry of CMA, Chinese Academy of Meteorological Sciences, Beijing 100081, China; yili@cma.gov.cn; 2Chinese Center for Disease Control and Prevention, Beijing 102206, China

**Keywords:** visceral leishmaniasis, meteorological factors, outbreak, gee, Kruskal–Wallis test

## Abstract

Although visceral leishmaniasis disease is controlled overall in China, it remains a serious public health problem and remains fundamentally uncontrolled in Jiashi County, Xinjiang Uygur Autonomous Region. During 2005–2015, there were two outbreaks in Jiashi County. Assessing the influence of meteorological factors on visceral leishmaniasis incidence is essential for its monitoring and control. In this study, we applied generalized estimating equations to assess the impact of meteorological factors on visceral leishmaniasis risk from 2005 to 2015. We also compared meteorological factors among years with Kruskal–Wallis test to explore possible reasons behind the two outbreaks that occurred during our study period. We found that temperature and relative humidity had very significant associations with visceral leishmaniasis risk and there were interactions between these factors. Increasing temperature or decreasing relative humidity could increase the risk of visceral leishmaniasis events. The outbreaks investigated might have been related to low relative humidity and high temperatures. Our findings will support the rationale for visceral leishmaniasis control in China.

## 1. Introduction

Leishmaniasis is considered one of the most neglected tropical diseases worldwide, yet this disease has received little attention and few resources despite its serious impacts on both economic development and quality of life [1]. Visceral leishmaniasis (VL), also known as kala-azar (KA), is a parasitic infectious disease that affects human health [2].

VL remains a serious public health threat in China [3]. Two epidemiological types (mountain type and desert type) of VL have been classified in western China, based on the ecosystem and source of infection [4,5]. The desert type is an anthroponotic disease that is endemic in the oases of the plains in Kashi Prefecture, Xinjiang Uygur Autonomous Region [5,6]. VL most frequently occurs in young people under 20 years of age [7]. In east China, the transmission cycle is from humans to humans, and in southwest China, it is from canines to humans. However, in Jiashi county, it is still unclear whether some kind of animal is the pathogen source [8].

Jiashi County in Kashi Prefecture has been one of the most-affected VL-endemic areas in China within the last century [9,10,11]. After a large-scale disease control program was conducted in endemic areas during the 1950s, VL was successfully controlled by 1958. After 20 years of control, no new local cases of VL were reported in endemic areas until 1983 [3,12]. However, compared with other areas of China, in which the prevalence of VL is well-controlled, the VL prevalence in Jiashi County has not decreased in the past 10 years [13]. Moreover, two outbreaks occurred during 2005–2015 and more than 95% cases occurred in children under 3 years of age. There were 393 cases reported during 2008–2009, which accounted for 39% of all cases reported in the whole country (393/1009). There were 494 VL cases reported during 2014–2015, representing about 62% of total cases in the country (494/798). Differing from the long-term trend in Jiashi, other areas of China have showed decreasing trends during 2005–2015 [14].

The female sand fly is the main vector of the Leishmania parasite. Phlebotomine sand flies are vectors of the Leishmania parasite in the Xinjiang Uygur Autonomous Region and *Phlebotomus wuii (P. wuii)* is the main Leishmania vector in Jiashi County [15,16]. The life cycle of *P. wuii* sand flies is influenced by both relative humidity (RH) and temperature, resulting in fluctuating population density of the vector. Thus, climate-related changes in the life-cycle dynamics of the Leishmania pathogen may increase or decrease the potential rate of transmission of disease [17,18,19].

In this study, we applied generalized estimating equations to study the associations between meteorological factors and VL events from 2005 to 2015. We also explored the reasons behind the outbreaks in Jiashi County by examining the annual and monthly characteristics of meteorological factors during the study period. Our study aimed to find the associations between meteorological factors and reported VL cases. Our findings will be important for determining the best time to intervene in the epidemiological chain of VL.

## 2. Materials and Methods

### 2.1. Study Area

Jiashi County is located in Kashi Prefecture, in the southwest part of the Xinjiang Uygur Autonomous Region. Jiashi County is within 39°16′–40°00′ N, 76°20′–78°00′ E, with a distance approximately 140 km from east to west and 80 km north to south, covering an area of 6667 km^2^ (Figure 1). Jiashi County has a typical temperate arid continental climate, with an annual mean temperature of approximately 11.7 °C, an annual maximum temperature of approximately 41.5 °C, and a minimum temperature of approximately −25.5 °C.

### 2.2. Data Collection 

Data of VL events reported from 2005 to 2015 were used in this study. The data were obtained from passive surveillance data reported through the web-based National Diseases Reporting Information System (NDRIS), operated by the Chinese Center for Disease Control and Prevention. In China, VL cases are compulsorily reported via the NDRIS according to the National Regulation on the Control of Communicable Diseases of 2005 [20]. Each record includes information such as name, age, sex, diagnosis, date of birth, date of onset, and current address code, among other factors.

Daily meteorological data (mean temperature, minimum temperature, maximum temperature, RH, precipitation, mean land surface temperature, minimum land surface temperature, maximum land surface temperature) were obtained from the China Meteorological Administration. From 1 January 2005 to 31 December 2015, data from a total of 4017 days were recorded, and a total of 1112 VL cases were reported.

### 2.3. Statistical Methods

A generalized estimating equation with first-order autocorrelation (AR1) was used to address possible serial correlation in the number of VL events. Monthly mean (temperature and relative humidity (RH)) or sum (precipitation) data were used, and models were implemented using the geepack package in R version 3.4 [21,22,23]. Covariates included ambient temperature, land surface temperature, number of days with mean ambient temperature in the range 21–28 °C, RH, and precipitation. Considering that VL usually involves an incubation period of 2–6 months, we used lagged data in the model; that is, the VL event month was regressed with monthly meteorological data from 2–6 months prior.

We compared the yearly meteorological factors using the Kruskal–Wallis or Mann–Whitney U-test, accordingly. The advantage of using the Kruskal–Wallis test in heterogeneity of variance cases is that the ranked data meet the normality assumption of a one-way analysis of variance. We considered *p* < 0.05 as significant in our statistical tests (all were two-sided). 

## 3. Results

### 3.1. Summary Statistics of VL Cases and All Meteorological Factors

Table 1 shows summary statistics of VL cases as well as meteorological factors. Considerable variation in VL cases and all meteorological factors could be seen and the number of VL cases displayed an “outbreak” mode; 75% of data values were zero and the maximum was as high as 55 (daily mean value). Also, it could be seen that temperature variables were widely distributed.

Figure 2 shows the correlation matrix of VL cases and meteorological factors. High correlation coefficient could be seen between RH and temperature, suggesting that these two types of variables could not be incorporated into one regression model. The correlation coefficient between RH and precipitation was less than 0.30, indicating a weak correlation; these two factors could be incorporated into a regression model without causing model instability. 

### 3.2. Monthly Characteristics

The monthly values of VL events and meteorological factors are shown in Figure 3. It could be seen that VL occurred mostly in autumn and winter and was less frequent in spring and summer. A reason for this might be that there is an incubation period of four months after *P. wuii* biting before a VL case attack. Figure 3 also shows significant seasonal characteristics of meteorological factors; the highest precipitation levels were in May and the lowest in November. For both ambient and land surface temperature, the highest values appeared in June and the lowest in January. As for RH, higher values were present in winter and lower values in spring.

### 3.3. Generalized Estimating Equations

In this study, we applied generalized estimating equations to assess the effects of meteorological factors on the risk of VL cases using monthly data, taking every year’s data as repeated observations. Table 2 shows the percentage change for the relative risk of VL with a one-unit increase in the four temperature variables (number of days per month with mean temperature 21–28 °C, mean temperature, minimum temperature, and minimum land surface temperature), using a single-variable meteorological factor model. It can be seen that all temperature variables had significant (*p* < 0.01) positive associations with VL events at different lag months, whereas RH had a significant (*p* < 0.01) negative association with VL events. The relative risk of a VL event showed a 3.41–6.50% and 4.85–11.80% increase with +1 days in the number of days with mean ambient temperature 21–28 °C and with a 1 °C increase in mean temperature at different lag months, respectively. The associations of minimum temperature and land surface minimum temperature were similar to that of mean temperature: The relative risk of a VL event had a 4.79–12.09% and 3.21–9.23% increase with a 1 °C increase in minimum temperature and land surface minimum temperature at different lag months, respectively. The relative risk of a VL event showed a 3.21–9.79% decrease, with a 1% increase in RH at different lag months. No significant associations were observed between monthly total precipitation and VL events.

As there was a strong correlation between temperature variables and RH, these two kinds of variable could not be input together into one model. To explore the interactive effects among these two kinds of variables, we set an interaction item in the model. Table 3 shows the interactive effects between RH and temperature variables when temperature was input into the model as the independent variable. Significant negative interactions were observed between the three temperature variables and RH at lag 3 to 6 months. As RH itself had a negative association with VL events, the negative interactions suggested that decreasing RH would enhance the effects of temperature. Compared with estimates from the single-variable model, the estimated coefficients of temperature variables with interaction were higher; the relative risk of a VL event showed a 26.45–33.37% increase, with +1 days in the number of days with a mean ambient temperature of 21–28 °C at different lag months. The relative risk of a VL event had a 31.00–35.45% increase, with a 1 °C increase in mean temperature at different lag months. The correlations of minimum temperature and land surface minimum temperature were similar to that of mean temperature; relative risk of a VL event showed a 31.11–48.49% and 28.15–43.49% increase with a 1 °C increase in minimum temperature and land surface minimum temperature at different lag months, respectively.

Table 4 shows the estimates of RH as the independent variable and interactions with temperature variables. Significant positive interactions between the three temperature variables and RH could be seen at 2 to 5 months. As temperature variables all had positive associations with VL events, the positive interactions suggested that increasing temperatures would enhance the effects of RH; the percent change in VL risk and RH were in the range 6.03–9.90%, slightly higher than that of the single-RH model. 

### 3.4. Differences in Meteorological Factors among Years

Table 5, Figure 4 and Figure 5 show the yearly values (mean or sum values) of VL events and meteorological factors during our study period; significant variations could be observed. The mean minimum land surface temperature ranged from 0.60 °C to 4.53 °C, precipitation ranged from 47.00 mm to 208.20 mm, and RH ranged from 47.31% to 55.82%. The variation in ambient temperature was relatively smaller; mean temperature was in the range of 11.28–13.33 °C. In some years, VL cases occurred sporadically, and there were two outbreak periods, in 2008–2009 and 2014–2015. The count of VL cases began in 2005 with a total of 114 cases, and then decreased each year in 2006 and 2007; however, from 2007, there was an increasing trend, with the first peak of more than 284 cases in 2008. The number of cases then decreased each year until reaching the lowest value (13 cases) in 2013. This number began to increase again, until a second peak was reached in 2015, which was also the largest peak with 381 cases. Four peaks of VL occurred in 2008, 2009, 2014, and 2015 during the research period: About 25.53% (284/1112) of the total cases occurred in 2008, 22.22% (247/1112) in 2009, 13.39% (149/1112) in 2014, and 34.26% (381/1112) in 2015.

To explore the possible reasons behind these outbreaks, we recombined our data into new datasets: Dataset 1 contained the years 2008 and 2009, dataset 2 contained 2014 and 2015, and dataset 3 contained years 2008, 2009, 2014, and 2015. Datasets 1–3 included data of “outbreak” years. We selected data of “non-outbreak” years and combined these into three additional datasets: Dataset 4 contained years 2005 to 2007, dataset 5 contained 2010 to 2013, and dataset 6 contained the years 2005–2007 and 2010–2013. The significant levels of the meteorological differences among these datasets were shown in Table 6.

Figure 5 shows the yearly density distributions of mean temperature and RH by year. It could be seen that the distributions of mean temperature among years were not different, whereas those of RH differed significantly.

## 4. Discussion

At one time, China was a severe VL-epidemic area. Since the 1960s, VL has been nearly eliminated in central and eastern parts of the country. However, VL is still endemic in some areas of western and southwestern China, such as the Xinjiang Uygur Autonomous Region, Gansu and Sichuan provinces. Among these areas, Jiashi County has experienced the most severe VL epidemic events in the past 20 years, with two outbreaks of VL in the county since 2005. During outbreak periods, more than half of all VL cases reported in China occurred in Jiashi. At the same time, VL showed a steady decreasing trend in other areas (Figure A1). Infectious disease outbreaks can be induced by a large migrant population entering an epidemic area, resulting in an increase in the non-immune population, or by a dramatic increase in the population number or density of the vector species, or prolonged growth and reproduction time, all of which are often owing to environmental damage or changes, such as those caused by deforestation. However, the reasons for the VL outbreaks in Jiashi County remain unclear. After the outbreak in 2008, the Jiashi government and Chinese Center for Disease Control and Prevention began to address the problem very seriously and initiated prevention and control measures including indoor residual spraying, health education, case surveillance, and timely treatment. However, these measures had very limited effect. Another VL outbreak occurred in 2009, with another more severe outbreak occurring in 2014–2015. During this time, the increase in the vector population (aged 0 to 3 years, which was the main age reported in VL cases) was inconsistent with that of VL events (Figure A2), suggesting that the measures taken were insufficiently effective and other important factors were not yet under control.

As for VL in In Jiashi County, *P. wuii* is the main vector of VL; therefore, the growth and reproduction of *P. wuii* are important factors in VL events. Apart from the natural environment, meteorological factors like temperature, precipitation, and RH have important impacts on the growth and reproduction of *P. wuii*. This species prefers a relatively dry, warm, and dark environment, with 21–28 °C believed to be the favored temperature range of *P. wuii*. Precipitation could affect the population density by changing the living environment of the vector; excessive precipitation would be unfavorable for *P. wuii* larvae. RH is another important factor, as high RH is unsuitable for *P. wuii*, such as in northwestern China where this species does not thrive.

In this study, we used generalized estimating equations to assess the associations between meteorological factors and VL events. Our results showed that temperature and RH had very significant impacts on VL events. Increasing temperature or decreasing RH would increase the relative risk of VL events, and there were significant interactions in the associations between temperature and RH. That is, lower RH could enhance the effect of high temperatures on VL risk, and high temperatures could enhance the effects of low RH on VL risk. However, higher RH could attenuate the effect of high temperatures, and lower temperatures could attenuate the effect of low RH. High temperatures and low RH suggest higher relative risk of VL, in comparison with high temperatures and high RH or low temperatures and low RH.

We also compared the meteorological factors between “non-outbreak” and “outbreak” periods, to further investigate the reasons behind the outbreaks. Our findings showed that there were relatively low RH and precipitation and relatively high temperatures during outbreak periods. Moreover, these factors did not act alone; interactions among them could be observed, as mentioned above. The interactions between temperature and RH might be causal factors during the two outbreak periods. The second outbreak was more severe and there might have been two reasons for this outbreak; first, the mean maximum land surface temperatures in outbreak years were higher than those in non-outbreak years; second, precipitation during outbreak years was much lower than that in non-outbreak years. Although we found no significant association between precipitation and VL events in the single-variable model with the total amount of precipitation, we did find some significant negative associations between precipitation and VL in the two-variable model with land surface maximum temperature (see the Table A1). 

We noted that although some of the lowest RH values appeared in 2007, the precipitation amount in that year was much higher than that in 2006 and 2008. Moreover, 2007 had the lowest annual mean surface temperature (0.60 °C) in our study period, much lower than those of other years. As the first three growth periods of *P. wuii* are in the soil near the land surface, land surface temperature would have an important impact on the growth of *P. wuii*. Another notable year was 2015, which had the third lowest RH and the highest ambient temperatures (mean, maximum, and minimum values) and minimum land surface temperatures of the study period; the largest peak in VL events occurred in 2015.

VL is a climate-sensitive disease owing to the preferred breeding conditions of sand fly vectors. Our finding that temperature and RH could affect the VL pattern is consistent with numerous findings regarding the effects of meteorological factors on disease as well as the life cycles of the sand fly and Leishmania [17,18,19,24,25,26,27]. 

However, findings regarding the geographic patterns of how climate could affect VL differed among these reports; differences might also exist among vector species. In Algeria, one study showed that both temperature and RH had positive associations with VL, but temperature had a greater effect [26]. VL prevalence in Bangladesh [28] was found to be positively associated with RH and precipitation and negatively associated with yearly average maximum temperature. In Colombia, VL events were found to have a positive link with temperature [29]. Researchers in Iran identified a significant positive association between VL and temperature, and a negative association between precipitation and VL. A study conducted in Tunisia [27] showed that average temperature, cumulative rainfall, and average RH, with 1–4 month lags, all had positive associations with VL events. In Brazil [30], only rainfall was found to have a positive association with VL. A report from India [31] indicated that VL had a positive association with rainfall and a negative association with RH. To sum up, the relationship between VL and climate differs according to geographic characteristics.

One strength of this work is that we investigated both ambient and surface land temperature because growth in sand flies takes place in both soil and air during different stages. We also explored the interactions between temperature and RH, to determine a possible explanation for the VL outbreaks in Jiashi County. Our findings suggest that these meteorological factors might act together to facilitate outbreaks of VL disease. One limitation of this study is that we lacked soil data; we therefore could not examine changes in the soil and vegetation conditions.

## 5. Conclusions

This work provides a foundation to explore the impacts of meteorological factors on VL. Our results indicated there are positive associations between temperature and VL risk, and negative associations between RH and VL risk. Further investigation of the relationships between changes in climate conditions and VL incidence is warranted. 

## Figures and Tables

**Figure 1 ijerph-16-01775-f001:**
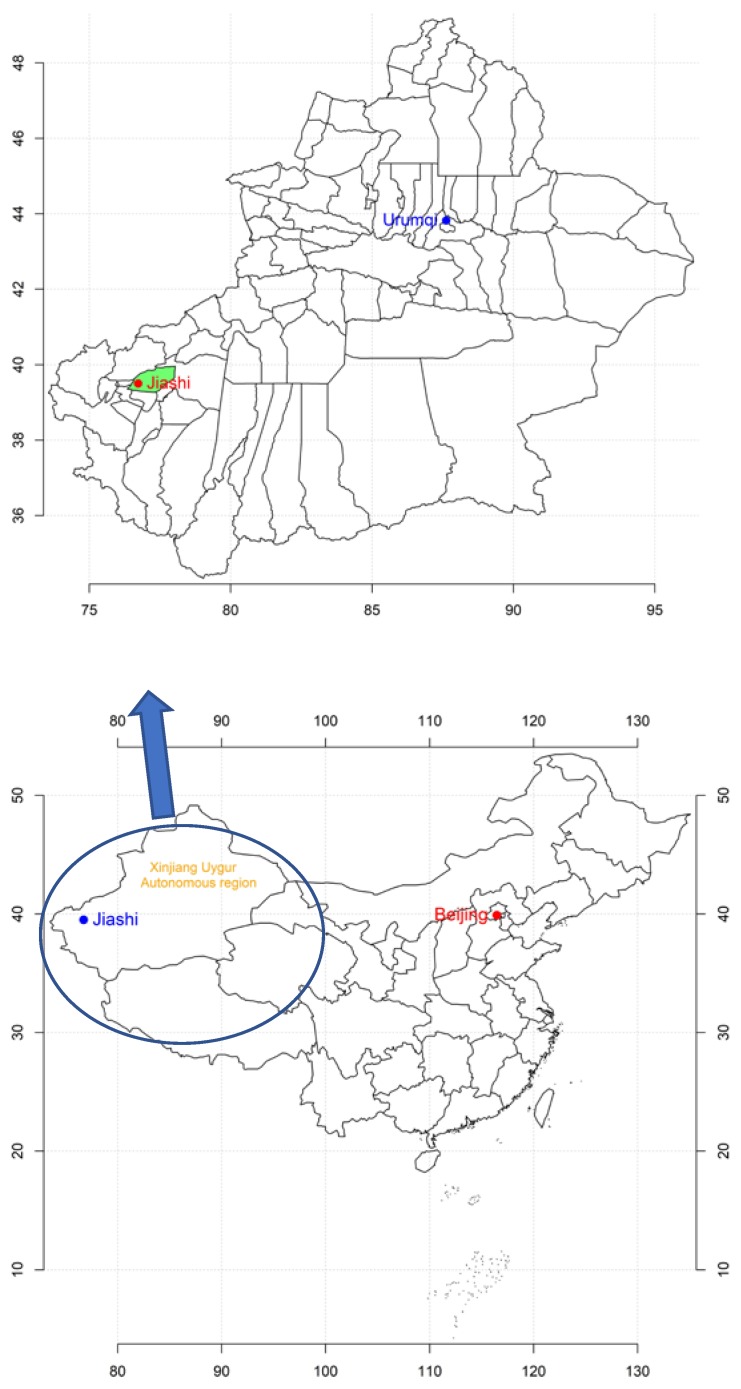
Study area. Blue circle indicates Xinjiang Uygur Autonomous Region; green area indicates Jiashi County.

**Figure 2 ijerph-16-01775-f002:**
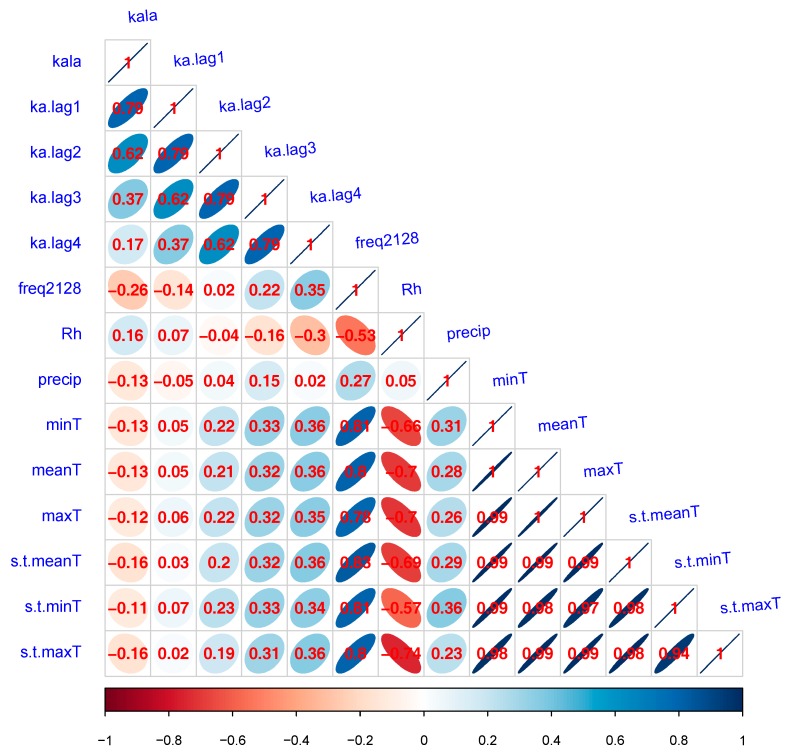
Correlation matrix of VL and meteorological factors. Size and color indicate the magnitude of positive or negative correlation coefficients.

**Figure 3 ijerph-16-01775-f003:**
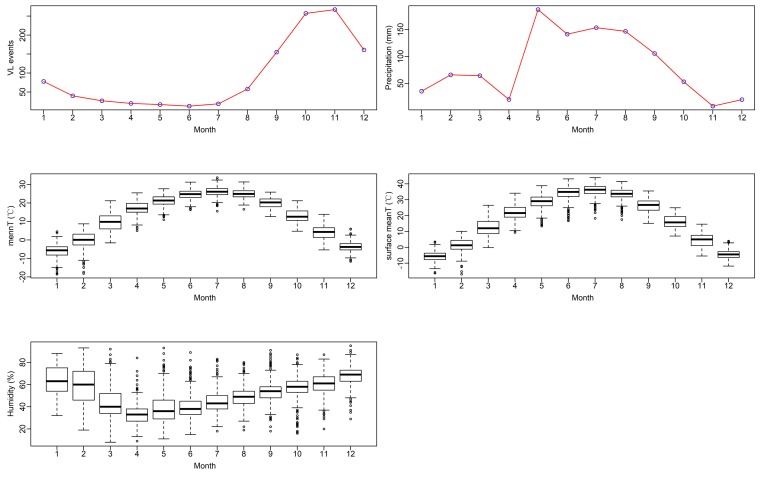
Monthly characteristics of VL events and meteorological factors. The first two panels are average values of monthly VL events and precipitation, and the other three panels are boxplots of monthly mean temperature, monthly mean surface temperature, and relative humidity.

**Figure 4 ijerph-16-01775-f004:**
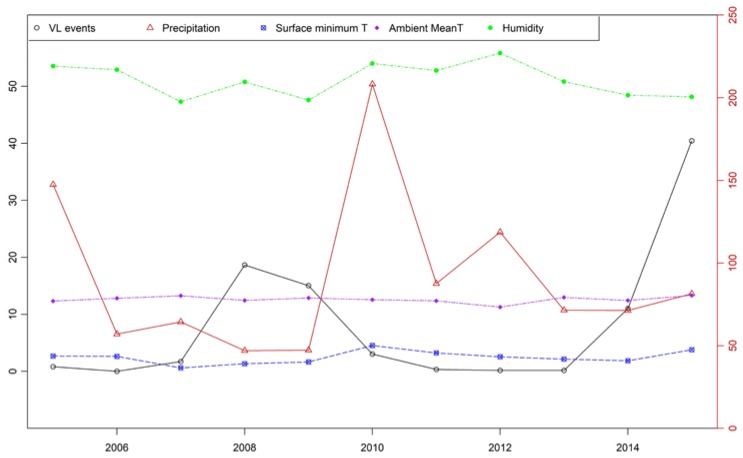
Yearly mean values of VL events, precipitation, surface land minimum temperature, and relative humidity.

**Figure 5 ijerph-16-01775-f005:**
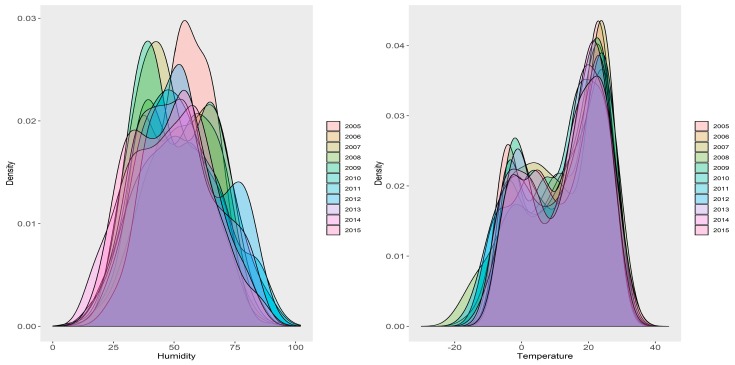
Yearly density distribution plots of ambient mean temperature and relative humidity. Density distribution of different years could be told by different colors.

**Table 1 ijerph-16-01775-t001:** Summary of visceral leishmaniasis (VL) cases and meteorological factors *.

Variables	Minimum	25%	Median	Mean	75%	Maximum
VL case	0	0	0	0.28	0	55
Mean temperature (°C)	−18.50	2.20	15.40	0.00	22.90	33.70
Relative humidity (%)	8.00	39.00	51.00	51.11	63.00	95.00
Precipitation (mm)	0.00	0.00	0.00	0.25	0.00	56.90
Mean land surface temperature (°C)	−16.90	3.05	19.10	17.11	30.70	43.90
Maximum temperature (°C)	−11.70	9.50	23.20	20.23	31.10	41.50
Minimum temperature (°C)	−25.50	−3.50	8.10	6.06	15.60	29.40
Maximum land surface temperature (°C)	−9.60	23.35	43.50	40.99	59.10	75.10
Minimum land surface temperature (°C)	−26.00	−7.50	3.90	30.70	12.70	26.20

* All data are daily mean values.

**Table 2 ijerph-16-01775-t002:** Percent change of VL events associated per unit increase of meteorological factors from single-variable models.

Meteorological Factors	Lag (Months)	Percentage(%)	*p*-Value
Number of days with mean ambient temperature in the range of 21–28 centigrade	2	0.0050 (−0.030, 0.040)	0.78
	3	0.042 (0.011, 0.073)	0.01 *
	4	0.069 (0.038, 0.099)	<0.01 *
	5	0.056 (0.030, 0.082)	<0.01 *
	6	0.035 (0.0046, 0.065)	0.02 *
Monthly mean temperature (°C)	2	0.053 (0.021, 0.085)	<0.01 *
	3	0.097 (0.052, 0.14)	<0.01 *
	4	0.12 (0.068, 0.17)	<0.01 *
	5	0.087 (0.055, 0.12)	<0.01 *
	6	0.050 (0.022, 0.077)	<0.01 *
Monthly total precipitation (mm)	2	0.0067 (−0.027, 0.041)	0.70
	3	0.020 (−0.005, 0.046)	0.12
	4	0.0040 (−0.021, 0.029)	0.76
	5	0.0049 (−0.021, 0.031)	0.71
	6	−0.023 (−0.051, 0.005)	0.12
Monthly mean relative humidity (%)	2	−0.007 (−0.026, 0.012)	0.48
	3	−0.031 (−0.052, −0.00)	0.00 *
	4	−0.064 (−0.086, −0.04)	<0.01 *
	5	−0.083 (−0.11, −0.05)	<0.01 *
	6	−0.099 (−0.13, −0.06)	<0.01 *
Monthly mean minimum temperature (°C)	2	0.058 (0.023, 0.093)	<0.01 *
	3	0.10 (0.056, 0.15)	<0.01 *
	4	0.12 (0.072, 0.17)	<0.01 *
	5	0.087 (0.055, 0.12)	<0.01 *
	6	0.049 (0.020, 0.078)	<0.01 *
Monthly mean minimum land surface temperature (°C)	2	0.056 (0.024, 0.089)	<0.01 *
	3	0.089 (0.050, 0.13)	<0.01 *
	4	0.095 (0.058, 0.13)	<0.01 *
	5	0.066 (0.040, 0.09)	<0.01 *
	6	0.034 (0.010, 0.057)	0.01 *

* Statistically significant.

**Table 3 ijerph-16-01775-t003:** Percent change of VL events associated per unit increase of temperature-related factors from multiple-variable models.

Meteorological Factors	Lag (Months)	Percentage (%)	*p*-Value	Estimate of the Interaction ^#^	*p*-Value of the Interaction *
Number of days with mean ambient temperature in the range of 21–28 centigrade	2	8.71 (−7.91, 25.32)	0.30	−0.002	0.24
	3	16.72 (5.07, 28.38)	<0.01 *	−0.003	0.023 *
	4	26.42 (19.13, 33.72)	<0.01 *	−0.005	<0.01 *
	5	29.03 (22.9, 35.13)	<0.01 *	−0.006	<0.01 *
	6	33.35 (25.4, 41.37)	<0.01 *	−0.008	<0.01 *
Monthly mean temperature (°C)	2	−2.25 (−16.15, 12.16)	0.76	0.002	0.25
	3	12.94 (1.76, 24.04)	0.02	−0.001	0.49
	4	25.73 (16.53, 35.02)	<0.01 *	−0.003	<0.01 *
	5	27.66 (24.21, 31.03)	<0.01 *	−0.004	<0.01 *
	6	28.57 (22.82, 34.14)	<0.01 *	−0.005	<0.01 *
Monthly mean minimum temperature (°C)	2	−1.22 (−23.45, 20.72)	0.91	0.002	0.48
	3	20.44 (4.17, 36.73)	0.01	−0.002	0.16
	4	38.45 (25.32, 51.61)	<0.01 *	−0.006	<0.01 *
	5	35.28 (28.03, 42.48)	<0.01 *	−0.006	<0.01 *
	6	31.14 (17.31, 44.93)	<0.01 *	−0.005	<0.01 *
Monthly mean minimum land surface temperature (°C)	2	6.05 (−15.56, 27.91)	0.59	0.000	0.99
	3	28.12 (11.32, 44.95)	<0.01 *	−0.004	0.02 *
	4	43.41 (31.61, 55.31)	<0.01 *	−0.007	<0.01 *
	5	28.46 (15.92, 40.93)	<0.01 *	−0.004	<0.01 *
	6	17.52 (−2.82, 37.98)	0.09	−0.003	0.19

^#^ Between variable and relative humidity. * Statistically significant.

**Table 4 ijerph-16-01775-t004:** Percent change of VL events associated per unit increase of relative humidity (RH) from multiple-variable models.

Variable Added in the Interaction in the Model	Lag (Months)	Percentage Change with One Percent Increase of Relative Humidity (RH)	*p*-Value	Estimate of the Interaction	*p*-Value of the Interaction *
Number of days with mean ambient temperature in the range of 21–28 centigrade	2	−1.71 (−5.16, 1.73)	0.33	0.000	0.58
	3	−3.43 (−7.05, 0.18)	0.06	0.000	0.64
	4	−6.22 (−9.73, −2.70)	<0.01 *	0.001	0.14
	5	−8.61 (−10.9, −6.29)	<0.01 *	0.000	0.34
	6	−9.74 (−12.5, −6.97)	<0.01 *	0.000	0.91
Monthly mean temperature	2	1.752 (−2.28, 5.79)	0.40	0.001	<0.01 *
	3	−2.45 (−6.23, 1.31)	0.20	0.002	<0.01 *
	4	−6.92 (−11.3, −2.48)	<0.01 *	0.002	<0.01 *
	5	−8.88 (−11.5, −6.18)	<0.01 *	0.001	0.01 *
	6	−9.83 (−12.3, −7.29)	<0.01 *	0.000	0.28
Monthly mean minimum temperature	2	2.313 (−1.98, 6.61)	0.29	0.002	<0.01 *
	3	−1.78 (−5.65, 2.08)	0.37	0.002	<0.01 *
	4	−6.14 (−10.5, −1.75)	0.01 *	0.002	<0.01 *
	5	−8.48 (−11.2, −5.73)	<0.01 *	0.001	<0.01 *
	6	−9.88 (−12.5, −7.24)	<0.01 *	0.000	0.43
Monthly mean minimum land surface temperature	2	2.008 (−2.29, 6.31)	0.36	0.001	<0.01 *
	3	−1.97 (−5.75, 1.81)	0.31	0.001	<0.01 *
	4	−6.03 (−1.87, −10.18)	<0.01 *	0.001	0.01 *
	5	−8.38 (−5.69, −11.07)	<0.01 *	0.001	0.03 *
	6	−9.90 (−7.15, −12.65)	<0.01 *	0.000	0.49

* Statistically significant.

**Table 5 ijerph-16-01775-t005:** Yearly values of VL events and meteorological factors.

Variables	2005	2006	2007	2008	2009	2010	2011	2012	2013	2014	2015
VL events ^#^	10	0	21	227	183	37	4	2	2	134	492
Precipitation (mm) #	147.50	57.00	64.40	47.00	47.30	208.20	87.60	118.70	71.50	71.30	81.40
Mean temperature (°C) *	12.34	12.82	13.26	12.45	12.88	12.56	12.37	11.28	12.98	12.44	13.33
Minimum temperature (°C) *	6.01	6.29	5.93	5.90	5.94	6.53	6.04	5.07	6.27	5.91	6.75
Maximum temperature (°C) *	19.45	20.18	21.46	20.02	20.83	20.09	19.68	18.81	20.88	20.05	21.05
Mean land surface temperature (°C) *	17.33	18.24	17.81	17.78	18.11	16.26	16.67	15.20	17.03	16.56	17.24
Mean value of minimum land surface (°C) *	2.68	2.63	*0.60*	1.33	1.65	4.53	3.23	2.55	2.15	1.86	*3.79*
Mean value of maximum land surface temperature (°C) *	38.84	41.05	42.24	40.92	42.77	37.83	40.39	38.39	43.79	42.94	41.77
Relative humidity (%)	53.55	52.91	47.31	50.78	47.60	54.00	52.77	55.82	50.84	48.45	48.15

^#^ Sum value; * Mean value.

**Table 6 ijerph-16-01775-t006:** Significant levels of meteorological differences among the five datasets.

Variables	2005–2007 Compared with 2008–2009	2010–2013 Compared with 2014–2015	Non-Outbreak Period ^#^ Compared with Outbreak Period ^§^	2008–2009 Compared with 2014–2015	2008 Compared with 2009	2014 Compared with 2015
VL events	<0.001 *	<0.001 *	<0.001 *	0.06	0.55	<0.001 *
Mean temperature	0.89	0.46	0.53	0.82	0.75	0.33
Maximum temperature	0.63	0.43	0.39	0.75	0.99	0.27
Minimum temperature	0.99	0.76	0.86	0.83	0.46	0.38
Relative humidity	<0.001 *	<0.001 *	<0.001 *	0.26	0.01 *	0.61
Mean land surface temperature	0.67	0.45	0.31	0.07	0.98	0.54
Mean value of maximum land surface temperature	0.10	0.04 *	0.01 *	0.80	0.40	0.37
Mean value of minimum land surface temperature	0.38	0.42	0.15	0.07	0.88	0.06
Precipitation	0.54	0.05 *	0.05 *	0.83	0.48	0.25

^#^ Non-outbreak period includes 2005–2007 and 2010–2013. ^§^ Outbreak period includes 2008–2009 and 2014–2015. * Statistically significant.

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
