# Peer review of "Associations between Meteorological Factors and Visceral Leishmaniasis Outbreaks in Jiashi County, Xinjiang Uygur Autonomous Region, China, 2005–2015"

_ijerph, 2019, doi:10.3390/ijerph16101775_

Round 1
Reviewer 1 Report
This paper presents a well-designed and conducted study to investigate meteorological factors associated with the re-emergence of visceral leishmaniasis in the Xinjiang Uygur Autonomous Region of China. The findings clearly point to independent association of low humidity and higher temperatures with re-emergence of the disease.
Figures 1 and 3 could be improved by indicating the measures portrayed in the x and y axes, and by including a more complete caption for Figure 3.
Author Response
Figures 1 and 3 could be improved by indicating the measures portrayed in the x and y axes, and by including a more complete caption for Figure 3.
Response: we added measures and names of X and Y axes in Fig 1 and Fig 3. Also we added some description in the caption of Fig 3 (in blue font).
Reviewer 2 Report
This manuscript reports some useful data regarding the impact of meteorological conditions on outbreaks of leishmaniasis in China. There are several issues that must be addressed before publication.
line 35: "...almost no animal host has been found (the percentage of possible animal host(s) found was below 0.3%..." It is unclear what the authors mean here. Is this the trap rate for a known reservoir species? the parasite infection rate of a known reservoir species? the percentage of trapped mammals of a known reservoir species? this needs clarification.
line 36 change "serious" to "affected"
line 42 "95% of cases occurred "in children" under 3 years of age
line 48 italicize "P. wuii" and all scientific names throughout manuscript
line 50 ...in fluctuating "population" densities (insert)
in Methods section
The ranked data of a K-W test do not meet the normality assumption of ANOVA; KW is a non-parametric test and does not assume normality as a parameter.
left justify the first column of all tables
All figures need titles that make the table stand alone even if taken out of the paper. They should describe the results, tests used, time when study was done and place. A complete description for interpreting the graphic is necessary, including what the different shapes and colors in reporting the results mean. If there is no specific meaning, then these extraneous factors should be removed from the tables.
Line 114. Not sure what the authors mean by "incubation time" for a sand fly. Does that mean time between feeding and ability to successfully transmit an infectious state of the parasite? I think the term the authors want to use is "extrinsic incubation period", though 4 months seems very long for this given that in the lab most leishmaniasis species have an extrinsic incubation period of 3 days.
Line 122 Effect on what? Number of cases? If so, you might want to use the term "correlation" in the title and in most places in the manuscript rather than "association". These words have very different meanings.
Most important issue: A confidence interval specifies a range of numbers, but your table indicates only one number under a couple of columns labeled 97.5% and 2.5%. This is very confusing and probably not correct. I think that what you intended to do is report the 2.5% value as the lower number in the CI and the 97% number as the upper number in the CI. Your first CI would be -7.94 - 25.37. If you are reporting something other than a CI, that needs to be explained.
line 204 "epidemic" or "endemic"?? I think you probably mean "endemic with occasional epidemics".
line 219 "vector population"? I think you probably mean "the mean age of cases".
Author Response
This manuscript reports some useful data regarding the impact of meteorological conditions on outbreaks of leishmaniasis in China. There are several issues that must be addressed before publication.
1. line 35: "...almost no animal host has been found (the percentage of possible animal host(s) found was below 0.3%..." It is unclear what the authors mean here. Is this the trap rate for a known reservoir species? the parasite infection rate of a known reservoir species? the percentage of trapped mammals of a known reservoir species? this needs clarification.
Response: we clarified here. The sentence was changed to “in east China, the transmission cycle is from human to human, and in southwest China, it is from canine to human. However, in Jiashi county, it is still unclear that if there has some kind of animal to be the pathogen source”. (Please see the blue font)
2. line 36 change "serious" to "affected"
Response: changed. (in blue font)
3. line 42 "95% of cases occurred " in children " under 3 years of age
Response: added. (in blue font)
4. line 48 italicize "P. wuii" and all scientific names throughout manuscript
Response: changed to italicize for all the manuscript. (in blue font)
5. line 50 ...in fluctuating "population" densities (insert)
Response: inserted (in blue font).
6. in Methods section. The ranked data of a K-W test do not meet the normality assumption of ANOVA; KW is a non-parametric test and does not assume normality as a parameter.
Response: here we used Kruskal-Wallis test (H-test) as an extension of the Wilcoxon test for large sample because meteorological data in this study was of more than three thousand days. We used this method to test the hypothesis that the temperature or relative humidity of different years originated from the same population. Kruskal-Wallis test is a non-parameter test and we think it is suitable for the meteorological data since we do not know the distributions of these data.
7. left justify the first column of all tables
Response: justified all the first columns of all tables. (in blue font).
8. All figures need titles that make the table stand alone even if taken out of the paper. They should describe the results, tests used, time when study was done and place. A complete description for interpreting the graphic is necessary, including what the different shapes and colors in reporting the results mean. If there is no specific meaning, then these extraneous factors should be removed from the tables.
Response: we added descriptions about tables and figures (in blue font). Please see line 100, lines 109-110, lines 120-122 and line 194.
9. Line 114. Not sure what the authors mean by "incubation time" for a sand fly. Does that mean time between feeding and ability to successfully transmit an infectious state of the parasite? I think the term the authors want to use is "extrinsic incubation period", though 4 months seems very long for this given that in the lab most leishmaniasis species have an extrinsic incubation period of 3 days.
Response: this “incubation time” meant the time between a bite by sand fly and a VL case reported. In Jiashi County, peak period of P.wuii occurrence was summer and autumn, while that of VL cases was autumn and winter. We clarified this point in the text in blue font.
10. Line 122 Effect on what? Number of cases? If so, you might want to use the term "correlation" in the title and in most places in the manuscript rather than "association". These words have very different meanings.
Response: we added words “on the risk of VL cases” in this line (in blue font).
11. Most important issue: A confidence interval specifies a range of numbers, but your table indicates only one number under a couple of columns labeled 97.5% and 2.5%. This is very confusing and probably not correct. I think that what you intended to do is report the 2.5% value as the lower number in the CI and the 97% number as the upper number in the CI. Your first CI would be -7.94 - 25.37. If you are reporting something other than a CI, that needs to be explained.
Response: we changed all the CI expression in all tables as “ 8.71 (-7.91, 25.32)” ( blue font in table2, table 3 and table 4).
12. line 204 "epidemic" or "endemic"?? I think you probably mean "endemic with occasional epidemics".
Response: changed to “endemic” ( blue font ).
13. line 219 "vector population"? I think you probably mean "the mean age of cases".
Response: more than 90% cases reported in Jiashi occurred in populations of under 3 year age. Here we clarified this point (in blue font).
Reviewer 2 Report
The authors have made a credible attempt to incorporate the suggested corrections provided by the reviewer. I believe the manuscript is ready for publication at this time.